# Poxviruses capture host genes by LINE-1 retrotransposition

**Sarah M Fixsen[1], Kelsey R Cone[1], Stephen A Goldstein[1], Thomas A Sasani[1], Aaron R Quinlan[1], Stefan Rothenburg[2], Nels C Elde[1,3]***

[1]Department of Human Genetics, University of Utah, Salt Lake City, United States; [2]Department of Medical Microbiology and Immunology, University of California, Davis, Davis, United States; [3]Howard Hughes Medical Institute, Chevy Chase, United States

**Abstract** Horizontal gene transfer (HGT) provides a major source of genetic variation. Many viruses, including poxviruses, encode genes with crucial functions directly gained by gene transfer from hosts. The mechanism of transfer to poxvirus genomes is unknown. Using genome analysis and experimental screens of infected cells, we discovered a central role for Long Interspersed Nuclear Element-1 retrotransposition in HGT to virus genomes. The process recapitulates processed pseudogene generation, but with host messenger RNA directed into virus genomes. Intriguingly, hallmark features of retrotransposition appear to favor virus adaption through rapid duplication of captured host genes on arrival. Our study reveals a previously unrecognized conduit of genetic traffic with fundamental implications for the evolution of many virus classes and their hosts.

## Editor's evaluation

This is a very important piece of work and a landmark in understanding the mechanism of horizontal gene transfer between viruses and their hosts. Horizontal gene transfer is a powerful mechanism of rapid evolution in many genomes, including humans and viruses. In this manuscript, a clever selection scheme is implemented to identify poxvirus genomes that have incorporated sequences from their host genome. Subsequent sequence analysis reveals that many of these sequences display the characteristics of terminal sequence repeats, splice junctions like mature mRNAs, and other features characteristics of the movement of LINE elements in cellular DNA genomes. Then, subsequent duplication of these transferred sequences is consistent with the viral DNA replication strategy and rationalizes many of the extant examples of horizontal gene transfer.

***For correspondence:**
nelde@genetics.utah.edu

## Introduction

The movement of genetic material from one organism to another by horizontal gene transfer (HGT) bypasses vertical inheritance from parent to offspring (*Keeling and Palmer, 2008*). HGT can shortcut more gradual mutational processes by providing the opportunity for adaptive leaps through single mutational events where genes can be transferred between genomes across kingdoms of life (*Keeling and Palmer, 2008*). HGT plays a central role in the diversification of many infectious microbes, which contend with complex environments and host immune defenses to persist and replicate. HGT is particularly well-described in bacteria, where it can contribute to the emerging crisis of multi-antibiotic resistant pathogens (*Wijayawardena et al., 2013*).

Many classes of viruses acquire genes from the hosts they infect by horizontal transfer (*Caprari et al., 2015*). A conspicuous example is Rous Sarcoma virus, in which recognition of the horizontal transfer of the host c-*Src* gene into the virus genome led to the discovery of oncogenes (*Swanstrom*

*et al., 1983*). Other co-opted host genes diverge to aid the virus in inhibiting host defenses (*Kawagishi-Kobayashi et al., 1997*; *Elde and Malik, 2009*). It has become increasingly clear that the acquisition of host genes is a common mechanism by which viruses of many classes gain adaptive advantages to propagate (*Dolja and Koonin, 2018*). Despite the prevalence of HGT in diverse viruses and its importance in virus evolution, the mechanisms by which genes are transferred from host to virus genomes are not well understood.

To investigate mechanisms of transfer we examined poxviruses, which encode a plethora of genes gained by horizontal transfer. Based on phylogenetic analysis, more than 25% of poxvirus genes were acquired by HGT from hosts (*Hughes and Friedman, 2005*), although similarities not evident in sequence comparisons but revealed by shared protein structures suggest that a higher proportion of genes were acquired by horizontal transfer (*Bahar et al., 2011*). Many captured genes adapt to act as inhibitors of host immune defenses to favor virus replication (*Elde and Malik, 2009*). Some act as host range factors because their deletion leads to restriction of infectible cell types or species, implying that new acquisitions may aid in host jumps (*Bratke et al., 2013*; *Haller et al., 2014*). While the most notorious poxvirus, variola virus which caused smallpox, has been eradicated, other extant poxviruses are capable of infecting humans and pose the risk of new pandemics, as appears to be the case for monkeypox (*Sklenovská and Van Ranst, 2018*). Poxviruses also serve as useful model systems for other viruses with large double-stranded (ds) DNA genomes (*Deeg et al., 2018*).

Because poxvirus-encoded copies of host genes lack introns, it was hypothesized that transfer occurs through a spliced mRNA intermediate (*Lefkowitz et al., 2006*). For integration, mRNA templates need to be reverse transcribed into virus genomes by either a retrovirus or a retrotransposon. The discovery of a provirus encoded in the genome of fowlpox virus (*Hertig et al., 1997*) and a Short Interspersed Nuclear Element (SINE) retrotransposon found in the taterapox virus genome (*Piskurek and Okada, 2007*) indicate that either is capable of catalyzing insertions in poxvirus genomes.

Given differences in biochemical activity, it is possible to detect patterns distinguishing mechanisms of retroviral versus retrotransposon-based transfer of host genes into virus genomes. Long Interspersed Nuclear Element-1 (LINE-1) retrotransposons are known to reverse transcribe and integrate host genes in a sequence independent manner, resulting in collections of processed pseudogenes in diverse host genomes (*Esnault et al., 2000*). LINE-1 transcripts encode two proteins: an RNA binding protein (ORF1) and an endonuclease (EN)/reverse transcriptase (RT) protein (ORF2). ORF2 preferentially nicks one strand of DNA at a TTTT/AA site in the host genome, and reverse transcription is primed from the exposed 3′hydroxl. LINE-1 elements can also integrate at sites of dsDNA breaks in an endonuclease-independent manner (*Morrish et al., 2002*). LINE-1 RT shows *cis*-preference, meaning it is most likely to transcribe the RNA from which it was translated, but SINE elements like the one found in the taterapox genome can hijack LINE-1 machinery to integrate following replication.

Sometimes LINE-1 enzymes promiscuously reverse transcribe host mRNA, which might facilitate transfers of host genes to virus genomes. It is plausible that the poly(A) tail of either the host mRNA or a poly(A) tract in SINE transcripts initializes LINE-1 reverse transcription, in addition to faithful recognition of LINE-1 transcripts. Short target site duplications (TSDs) of sequences ~8–20 bp flank LINE-1 insertions of either LINE-1 or host sequences, likely a product of repairing staggered breaks during integration. The resulting LINE-1-mediated flanking duplications are copies of host sequences as opposed to specific retrovirus sequences. Additional evidence of LINE-1 action comes from comparisons with closely related species lacking the insertion and instead containing what is termed an 'empty site'. Taken together, these features can help distinguish whether LINE-1 is involved in the transfer of host genes to virus genomes.

However, given that viruses rapidly mutate, unique signatures of retrovirus or LINE-1-mediated horizontal transfer may quickly degrade and obscure mechanisms of transfer. Therefore, we devised an experimental system to screen for the horizontal transfer of a gene from host to virus genomes using an artificial selection scheme. Combined with observations of viruses encoding recently acquired and highly conserved genes, we reveal LINE-1 elements as molecular accomplices in the transfer of host genes to virus genomes.

## Results

We hypothesized that examination of genes recently acquired by HGT might provide clues for revealing mechanisms of gene transfer. Virus homologs of the host Golgi Anti-Apoptotic Protein (GAAP; also

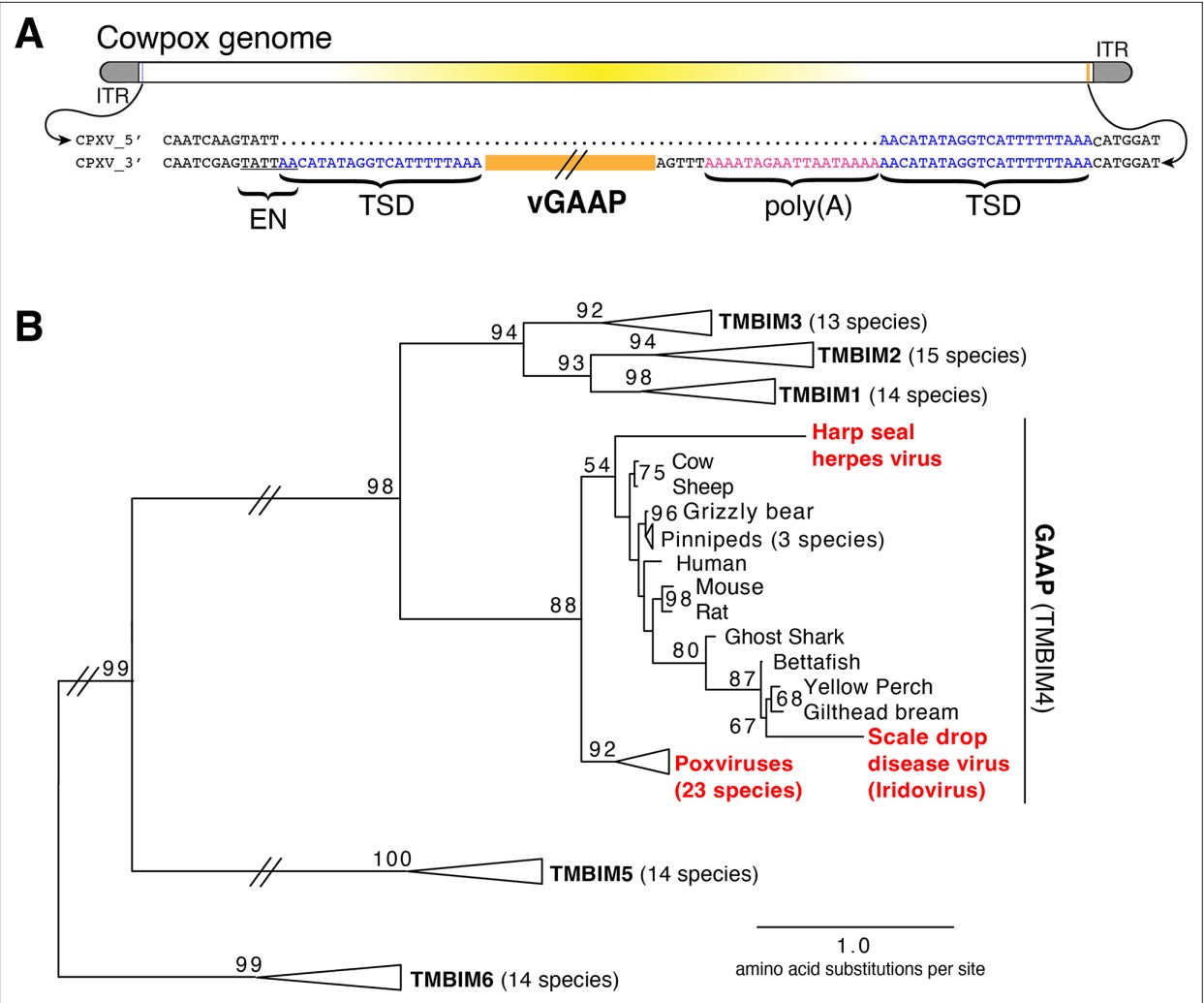

**Figure 1.** Retrotransposon-mediated transfer of GAAP into poxvirus genomes. (**A**) The poxvirus gene vGAAP (orange) is encoded near the inverted terminal repeat (ITR; gray) of cowpox genomes. Signatures of retrotransposition include TSDs (blue), a LINE-1-like endonuclease site (EN; underlined), and a partially degraded poly(A) tail (pink). Some cowpox genomes (CPXV_5′; top line) contain a pseudo-empty site, with a single copy of the TSD sequence. (**B**) Phylogeny of TMBIM proteins, including TMBIM4, or GAAP, and virus-encoded TMBIM4s (red). Bootstrap values >50 are indicated. TSD, target site duplication.

The online version of this article includes the following figure supplement(s) for figure 1:

**Figure supplement 1.** Genetic features of vGAAP in poxvirus genomes.

called transmembrane Bax inhibitor motif protein family 4 or TMBIM4), found in several orthopoxviruses, are ~75% identical to human GAAP (*Gubser et al., 2007*; *Saraiva et al., 2013*), suggesting a recent acquisition and/or highly conserved function favoring virus replication. Strikingly, we observed nearly identical 21 base pair sequences flanking the open reading frame of virus GAAP (vGAAP), indicative of a TSD, consistent with gene capture resulting from LINE-1-mediated retrotransposition of host mRNA (*Figure 1A* and *Figure 1—figure supplement 1*). The vGAAP gene is situated near one of two identical inverted terminal repeat (ITR) regions of the virus genome, and on opposite ends of the genome in various viruses, suggesting that it was originally transferred into the ITR. A single copy of the putative 21 bp TSD at an analogous position near the other ITR in the cowpox genome suggests a pseudo-empty site (*Figure 1A*). Taken together, with a published account of a non-autonomous SINE in the genome of taterapox virus (*Piskurek and Okada, 2007*), these observations suggest that host genes can be retrotransposed into poxvirus genomes by LINE-1 activity in a process akin to the generation of processed pseudogenes.

Phylogenetic analysis revealed a single origin for the horizontal transfer of GAAP into poxvirus genomes. In addition, at least two other classes of DNA viruses, herpesviruses and iridoviruses, independently acquired GAAP by horizontal transfer (*Figure 1B*; *Bellehumeur et al., 2016*; *de Groof et al., 2015*). Another study of Squirrel Monkey cytomegalovirus revealed that the S1 gene encodes a protein more than 95% identical to squirrel monkey Signaling lymphocytic activation molecule family 6 (SLAMF6), followed by portions of the 3′ untranslated region (UTR) of the host mRNA (*Pérez-Carmona et al., 2015*). Taken together, these findings suggest that many DNA viruses acquire host genes by retrotransposition. However, the majority of host to virus gene transfers lack signatures of retrotransposition, which can quickly degrade through mutation, motivating our development of an experimental strategy for capturing HGT events in real time.

We developed a selection scheme using vaccinia virus (Copenhagen strain), which encodes nearly 200 genes in a 191,737 bp genome (See Materials and methods). We focused on K3L, a gene that

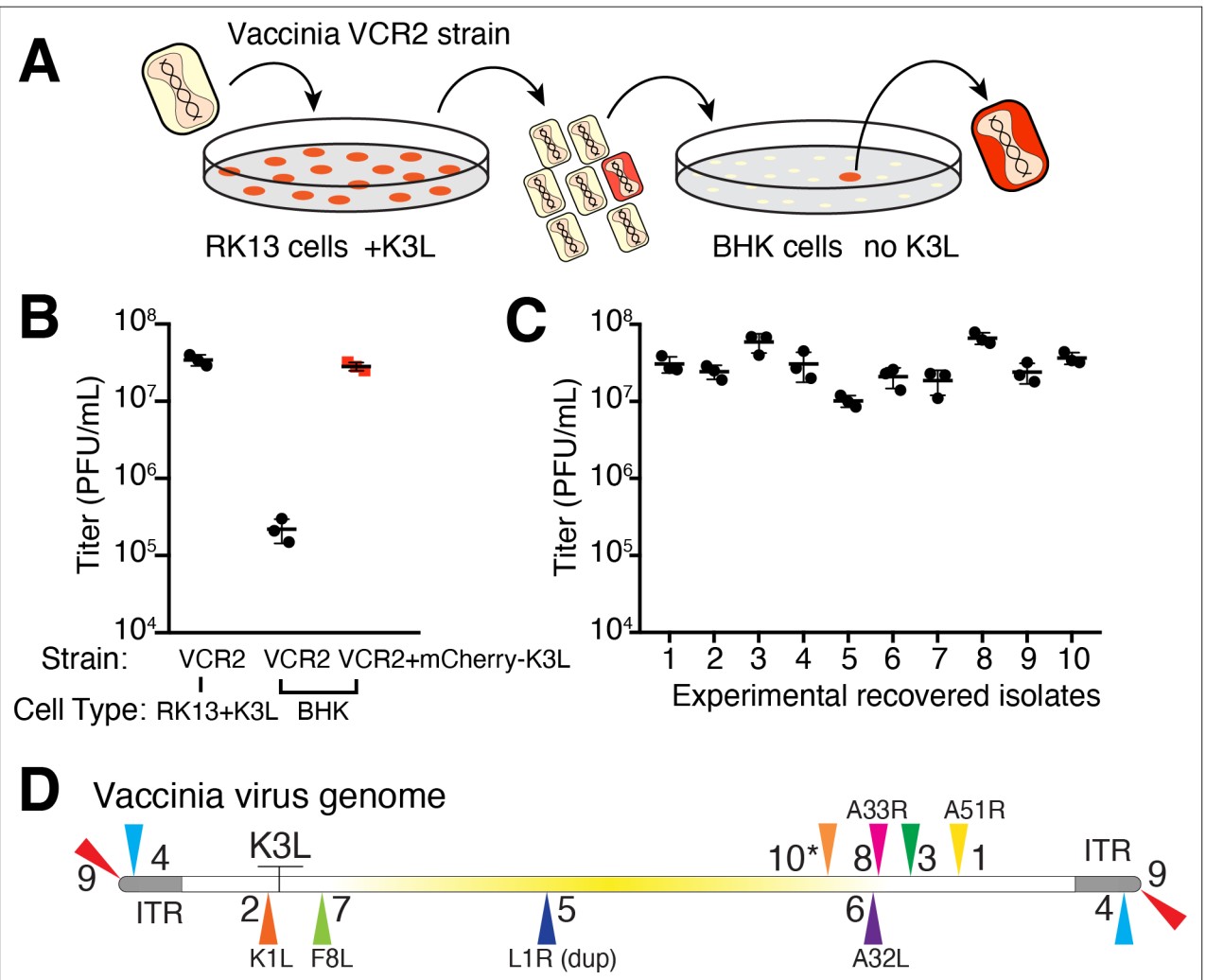

**Figure 2.** Experimental capture of K3L by horizontal gene transfer. (**A**) Viruses lacking K3L (VCR2) replicated in RK13 cells expressing mCherry-K3L and were transferred to BHK cells lacking K3L to select for capture of mCherry-K3L. (**B**) Replication of VCR2 in RK13-K3L cells and BHK cells compared to VCR2+mCherry-K3L (see Materials and methods). (**C**) Replication of recovered isolates in BHK cells. Three biological replicates of each strain/isolate with mean titer and standard error bars are shown in (**B**) and (**C**). (**D**) Vaccinia genome illustrating K3L integrations in recovered isolates indicated by colored triangles. Endogenous K3L location is shown. The central region of the genome is highlighted in yellow. Triangles above and below the genome denote positive and negative sense orientation, respectively. Virus genes interrupted by K3L integrations are indicated. The asterisk denotes a genomic rearrangement in isolate 10 (see *Figure 3A*).

The online version of this article includes the following figure supplement(s) for figure 2:

**Figure supplement 1.** Protein Kinase R (PKR) is activated by binding dsRNA which is made in the cytoplasm during a viral infection.

was presumably acquired from a host by an ancient poxvirus based on sequence identity with human eukaryotic initiation factor 2α (eIF2α; 27.6% identical) (*Beattie et al., 1991*; *Elde et al., 2009*; *Elde and Malik, 2009*). K3L diverged from eIF2α to encode a protein that blocks the host defense Protein Kinase R (PKR) pathway. PKR is activated by dsRNA produced by viruses in the cytoplasm during transcription of virus genes and virus replication. Upon activation, PKR phosphorylates eIF2α, leading to a block in protein translation and virus replication. K3L acts as a pseudo-substrate of PKR to competitively inhibit PKR from binding eIF2α (*Figure 2—figure supplement 1*; *Beattie et al., 1991*; *Kawagishi-Kobayashi et al., 1997*), and can be essential for productive infections depending on the host (*Langland and Jacobs, 2002*; *Park et al., 2019*; *Park et al., 2021*).

Using a mutant virus deleted for K3L and another PKR inhibitor, E3L (*Brennan et al., 2014*), we infected a rabbit kidney cell line (RK13) expressing chromosome-integrated copies of K3L fused to the gene encoding mCherry fluorescent protein (RK13-K3L) (*Figure 2A* and see Materials and methods for details). The host copy of mCherry-K3L complements the mutant virus lacking K3L, allowing it to replicate. Viruses grown in RK13-K3L cells were transferred to a non-complementing baby hamster kidney (BHK) cell line. Viruses lacking K3L replicate poorly in BHK cells, providing strong selection for viruses that 'reclaimed' the mCherry-K3L gene by horizontal transfer and/or somehow adapted in the absence of K3L to block host PKR and regain productive replication.

Using this scheme, we screened ~500 million viruses over many experiments and recovered ten virus isolates from mCherry-positive infected cell clones. In the restrictive BHK cells, viruses encoding K3L replicate ~100-fold better than the parent virus deleted for K3L (*Figure 2B*), allowing us to identify viruses with improved replication compared to the larger population of K3L-deficient viruses. After plaque purification of candidate mCherry positive isolates, we infected BHK cells with each recovered virus isolate and observed titers on par with viruses encoding K3L (*Figure 2C*). By sequencing the genome of each recovered virus isolate, we found that the mCherry-K3L fusion gene was integrated into the virus genome at a unique position in every isolate (*Figures 2D and 3A*), providing the opportunity to compare each independent integration event.

All integrations were consistent with a mechanism of LINE-1 retrotransposition-mediated HGT. The sequence of each transferred gene revealed hallmark features of mRNA gene capture, including 5' and 3' UTRs, precise intron splicing, and poly(A) tails (*Figure 3B*). For 7 of 10 isolates, TSDs flanked mCherry-K3L insertions and contained cut site sequences matching the consensus (TTTT/AA) of LINE-1 endonuclease (*Figure 3* and *Figure 3—figure supplement 1*). Although there were no TSDs flanking mCherry-K3L in isolates 5, 9, and 10, these integrations might result from an endonuclease-independent LINE-1 mechanism at DNA breaks (*Morrish et al., 2002*). Because our experiments lack any exogenous reverse transcriptase, we tested for endogenous activity in the RK13 cells and detected robust LINE-1 activity (*Figure 3—figure supplement 2*). These results suggest that LINE-1-mediated retrotransposition may be a common mechanism of HGT in poxvirus genomes. While we do not exclude the possibility that genes can also be transposed into the viral genome by retroviruses, this would require co-infection. LINE-1-mediated retrotransposition may be more likely simply because LINE-1 enzymes are nearly ubiquitous in mammals.

In general, poxvirus genomes are arranged with the essential genes, shared by all poxviruses, in the central region of the genome. Clade- or species-specific genes, which more commonly exhibit evidence of HGT, are enriched near the ends of the genome (*Lefkowitz et al., 2006*; *McLysaght et al., 2003*). This pattern may reflect that gene transfer into the central genomic region genome is more likely to interrupt genes vital for virus replication (*Lefkowitz et al., 2006*). Consistent with this, all but 1 of the 10 isolates, we recovered integrated K3L outside a roughly defined central region of the genome containing 90 core chordopoxvirus genes (*Lefkowitz et al., 2006*; *Figure 2D*). In the case of isolate 5, the K3L integration involved a duplication of two essential genes, G9R and L1R (*Figure 3B*). In a contemporaneous study (*Rahman et al., 2022*), virus genomes that acquired a host gene by horizontal transfer within a duplicated gene also propagated. In both cases, gene functions were preserved by duplication. Spurious gene duplications can appear frequently in poxvirus genomes (*Hughes and Friedman, 2005*), pointing to a flexible means of poxvirus adaptation involving horizontal transfer even into essential genes.

Previous studies found that duplication of poxvirus genes, followed by rapid gene copy amplification, can boost virus replication (*Brennan et al., 2014*; *Elde et al., 2012*). In our analysis, we discovered two isolates where K3L had undergone polymorphic copy number increases (isolates 4

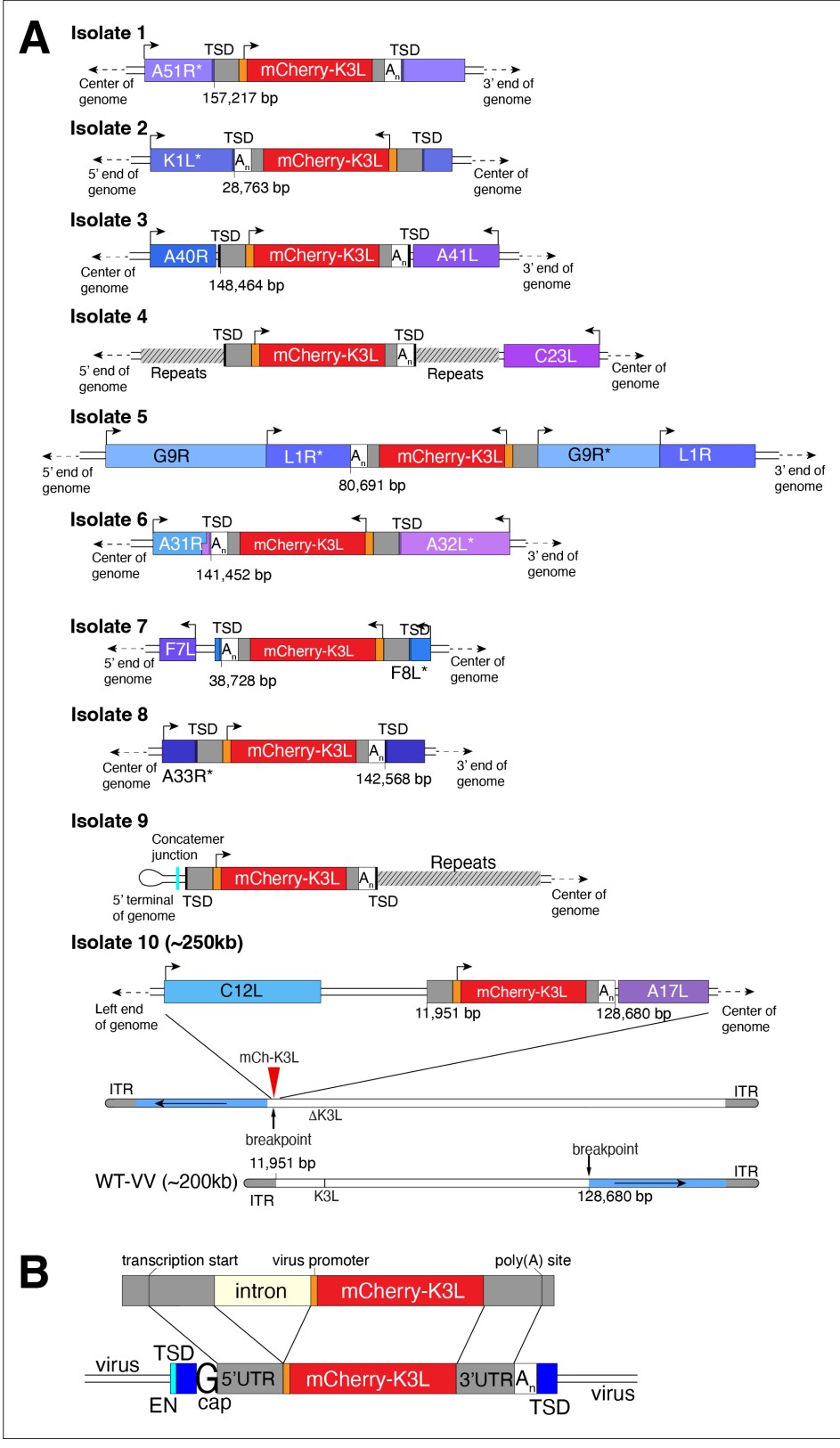

**Figure 3.** Each virus isolate exhibits signatures of LINE-1-mediated retrotransposition. (**A**) A detailed view of the region K3L inserted in each experimentally captured virus isolate. Arrows above cartoons indicate reading frame orientation. Flanking and/or interrupted (*) viral genes (blue/purple boxes), 3'/5' untranslated regions (UTRs; gray), and poly(A) tails (A_n; white) are shown. Genomic rearrangements included in isolate 10 are also shown.

*Figure 3 continued on next page*

*Figure 3 continued*

(**B**) Schematic of construct integrated in RK13-K3L cells (top) compared to K3L integrations in recovered viruses (bottom). Shared features include spliced introns, 5' and 3' UTRs, and poly(A) tails ($A_n$). Seven isolates have target site duplications (TSDs, blue) with LINE-1 endonuclease cut sites (EN, cyan). Six isolates encode a guanine (G cap) adjacent to the 5'UTR, indicative of 7-methylguanylate mRNA capping. LINE-1, Long Interspersed Nuclear Element-1.

The online version of this article includes the following figure supplement(s) for figure 3:

**Figure supplement 1.** Details of the insertions for each virus isolate that acquired K3L.

**Figure supplement 2.** An Alu-Neo reporter (cartoon above) was used to measure endogenous LINE-1 activity of the RK13-K3L cells in comparison to HeLa-HA cells.

and 5; *Figure 4A*). Homologous recombination, including unequal crossovers leading to duplications, requires as little as 16 bp of homology in poxviruses (*Yao and Evans, 2001*), suggesting that LINE-1 induced TSDs might facilitate gene amplification. Genes transferred within the ITR are duplicated on the other end by replication-based mechanisms (*McFadden and Dales, 1979*), as we observed in isolates 4 and 9 (*Figure 2C*). Genes integrated near the termini of the genome by HGT, like isolate 4, may be especially prone to duplication due to stretches of short, high copy tandem repeats found in and near the ITR regions (*Figure 4A* and *Figure 5—figure supplement 1A*). These observations are consistent with the idea that HGT occurs evenly across poxvirus genomes, but with an enrichment of transferred genes persisting near the ends, where insults to essential genes are less probable and duplication-based adaptation is favored (*Figure 5—figure supplement 1*). Along these lines, gene

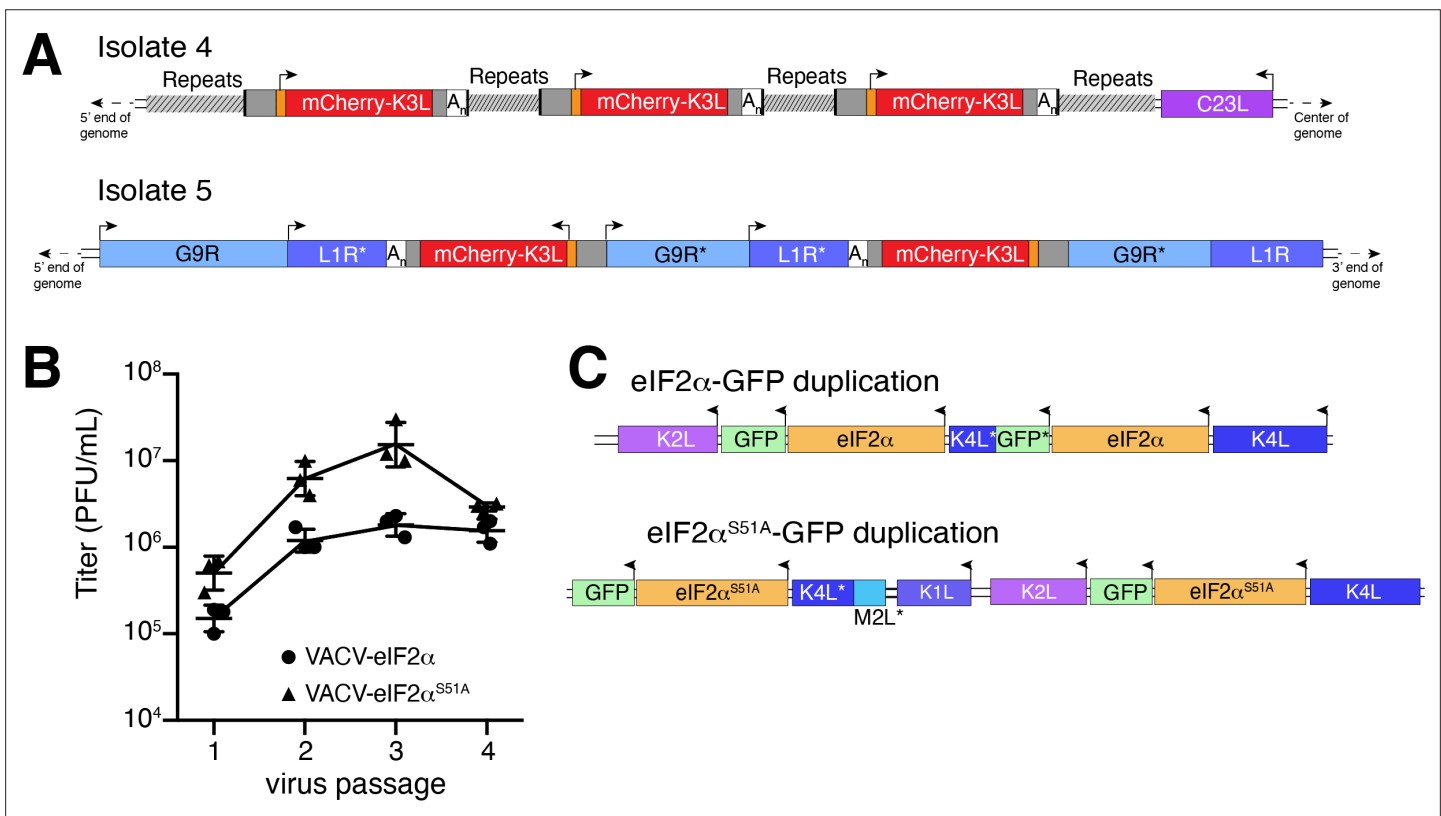

**Figure 4.** Homology-driven gene duplication of K3L and eIF2α. (**A**) Schematics of mCherry-K3L duplications in isolates 4 and 5. (**B**) Virus titers after serial infections of eIF2α viruses in HeLa cells (see Materials and methods). (**C**) Schematics of eIF2α duplications following virus passaging. Asterisks denote gene truncations.

The online version of this article includes the following figure supplement(s) for figure 4:

**Figure supplement 1.** Diagrams of the genomic region encompassing K3L in wildtype (WT) vaccinia virus, VCR2 strain, and viruses engineered to express eIF2α and eIF2α S51A.

families found only in specific lineages of poxviruses, indicating more recent acquisition, are more likely to be present in multiple copies than single-copy host-acquired genes found among diverse poxviruses (*Hughes and Friedman, 2005*).

While our experimental system revealed LINE-1-driven mechanisms of HGT, the selection scheme involved viruses regaining K3L, an existing virus gene as opposed to gaining a bona fide host gene. To better model virus adaptation following acquisition of a host gene, we engineered viruses with the human eIF2α gene in place of K3L (*Figure 4—figure supplement 1*; see Materials and methods). This recombinant virus better approximates the original ancient HGTs of the eIF2α gene that then evolved to become K3L in diverse clades of modern poxviruses (*Kawagishi-Kobayashi et al., 2000*). Because the viruses we engineered to encode eIF2α in place of K3L also lack E3L, which encodes another inhibitor of the host PKR antiviral pathway, experiments with these viruses focus strong selective pressure on eIF2α to adapt to inhibit PKR.

We performed four serial infections of viruses expressing eIF2α or an eIF2α variant lacking the PKR phosphorylation site (S51A) in RK13 cells. While the recombinant viruses replicate weakly, we observed a nearly tenfold increase in virus replication for both eIF2α-encoding viruses after passaging (*Figure 4B*). By sequencing the eIF2α region in evolved virus isolates, we found independent gene copy increases of eIF2α in both viruses (*Figure 4C*). These results support a critical role for gene copy increases following HGT in poxvirus evolution, as also observed for K3L and other genes (*Brennan et al., 2014*; *Elde et al., 2012*) and suggest a common pathway facilitating adaptation of host genes captured by poxviruses.

## Discussion

HGT is an important process facilitating virus evolution (*Caprari et al., 2015*; *Deeg et al., 2018*; *Hughes and Friedman, 2005*; *Koonin and Yutin, 2019*; *McLysaght et al., 2003*). Our discovery of LINE-1 retrotransposition-mediated HGT from host to poxvirus genomes has several notable implications. In addition to self-propagation in host genomes, retrotransposition to virus genomes could

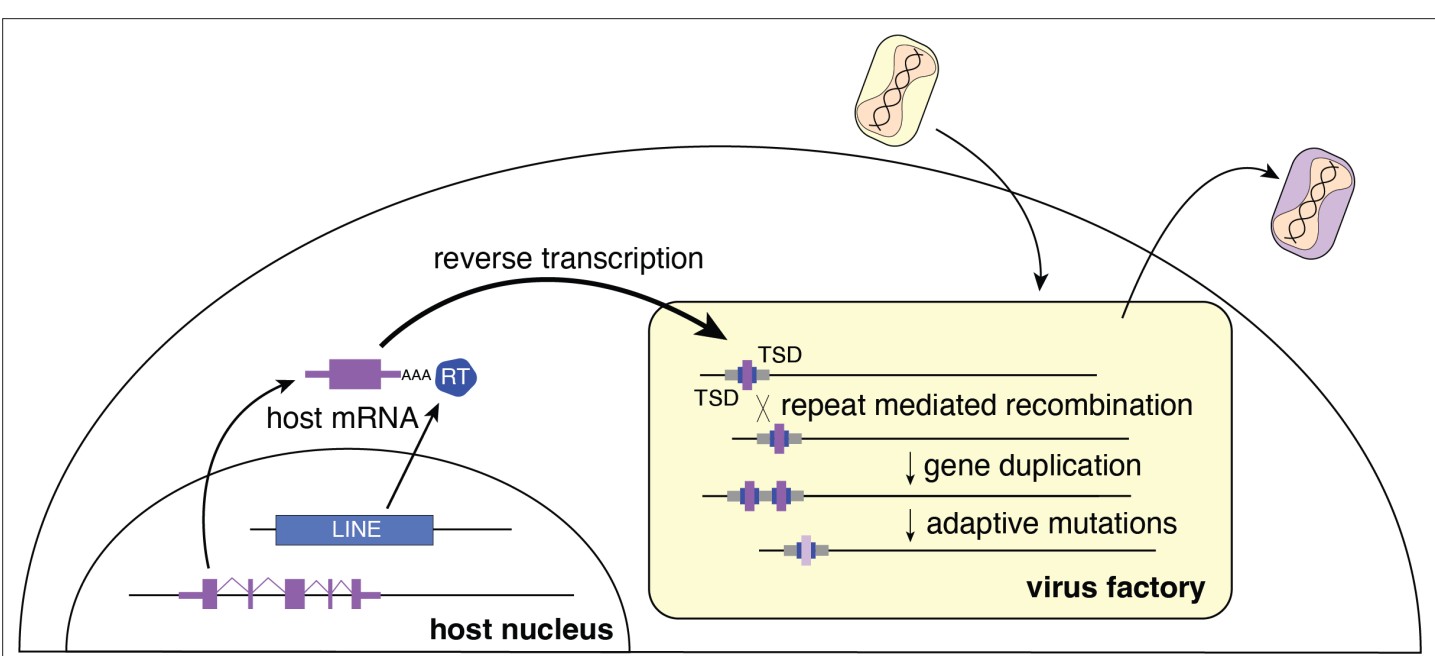

**Figure 5.** Model for horizontal transfer of host genes to poxvirus genomes. The schematic highlights how host genes (purple) are retrotransposed by LINE-1 reverse transcriptase (RT, blue), into virus genomes (yellow box). Following horizontal transfer, gene duplication is facilitated by unequal crossover recombination between TSDs (blue) or flanking repeat sequence (gray). LINE-1, Long Interspersed Nuclear Element-1; TSD, target site duplication.

The online version of this article includes the following figure supplement(s) for figure 5:

**Figure supplement 1.** Models of evolution of newly acquired virus genes.

greatly extend the range of LINE-1 or related elements that subsequently mobilize to a newly infected virus host (*Piskurek and Okada, 2007*). The extensive host range of some poxviruses, for example, between rodent and primate species, suggests the possibility of cross-host lineage transfer of LINE-1s, non-autonomous SINEs, and host transcripts between mammals. A process involving the frequent appearance of DNA transposons in some insect viruses suggests an analogous mechanism for the spread of transposable elements between insect species (*Gilbert et al., 2016*). In cases of retrotransposition of host transcripts, the resulting processed pseudogene products are enriched for highly expressed genes (*Zhang et al., 2004*). The overrepresentation of certain transcripts might provide clues for understanding and predicting how viruses might adapt with repurposed genes mimicking host functions (*Elde and Malik, 2009*). Currently recognized cases of HGT involving oncogenes, regulators of translation, like eIF2α, and antiviral genes that are highly expressed in response to infection. These examples of transcripts exhibit high gene expression and encode cellular functions easily repurposed to promote virus replication.

Repetitive TSDs generated by LINE-1 activity may also enhance poxvirus adaptation by facilitating rapid gene copy number amplification of horizontally transferred genes on arrival. Increases in gene copy number augment the probability of beneficial mutations appearing, allowing for swift diversification of newly acquired genes (*Elde et al., 2012*). Thus, form follows function as the repetitive termini of poxvirus genomes further enable the emergence and persistence of acquired genes predominately at the ends of the genome (*Figure 5*). As beneficial virus gene variants endure, they end up closer to the center of the genome as subsequently captured genes appear near the ends.

Similar mechanisms of virus adaptation by HGT may be shared by other diverse classes of nucleocytoplasmic DNA viruses (NCLDVs), in addition to poxviruses (*Deeg et al., 2018*; *Koonin and Yutin, 2019*). Many giant viruses, which can exceed a megabase in genome size and infect diverse host species, encode a variety of genes acquired by HGT, including ribosomal genes. Given a primary role for LINE-1 elements in facilitating gene transfer, we speculate that other classes of transposable elements may spur NCDLV evolution by mobilizing host genes into virus genomes across diverse ecosystems.

## Materials and methods

### Phylogenetic analysis and sequence comparisons of vGAAP and host GAAP/TMBIM4

TMBIM sequences for various species were downloaded from NCBI (Supplemental data) and aligned using the COBALT Multiple Alignment Tool. Amino acid Phylip alignments (see supplemental materials) were analyzed by PhyML (http://www.atgc-montpellier.fr/phyml/) for tree building with 100 bootstraps (*Guindon et al., 2010*).

### Cell culture

BHK (ATCC) and HeLa (gift from Adam Geballe, Fred Hutchinson Cancer Research Center) cells were cultured in Dulbecco minimum essential medium (HyClone) supplemented with 10% fetal bovine serum (FBS), 2 mM L-glutamine, and 100 µg/ml of pen/strep. RK13 cells were cultured in MEM-alpha supplemented with 10% fetal bovine serum (Gibco), 2 mM L-glutamine (HyClone), and 100 µg/ml of penicillin-streptomycin (pen/strep; HyClone).

RK13 cells expressing SLP-mCherry-K3L (RK13-K3L) were generated by transfecting $3 \times 10^5$ RK13-E3K3 cells (*Brennan et al., 2014*) with 2.5 µg piggybac vector (see supplemental sequences) and 0.5 µg transposase vector (gift from Ed Grow, University of Utah) with 9 µL Fugene-HD. Cells with the piggybac construct integrated were selected by the presence of 1–2 µg/ml puromycin, and populations were monitored for mCherry expression. After 2 weeks of selection, remaining polyclonal cells were propagated in MEM-alpha media supplemented with 10% FBS, 2 mM L-glutamine, and 100 µg/ml of pen/strep. All cultures were maintained, and infections were performed at 37°C in a humidified 5% $CO_2$ incubator.

### Virus strains

VC-2 refers to the Vaccinia virus-Copenhagen strain (*Goebel et al., 1990*) and VCR2 is a genetically modified isolate of VC-2 deleted for E3L and K3L (*Brennan et al., 2014*). To generate VCR2+mCherry-K3L,

we cloned pBlue_165_mCherry-K3L (see supplemental sequences) into the MCS of pBluescriptIIKS (−) (Addgene). This plasmid was transfected into BHK cells, which were then infected with VCR2 virus. Recombinant viruses were plaque purified in BHK cells and checked for purity by PCR amplifying across the mCherry-K3L insertion site. The amplicon was gel purified and sequenced.

To make eIF2α viruses, sequences flanking K3L from VACV were amplified from VC-2 viral DNA: 680 bp of 5′ homologous sequence was amplified with the primers K2Lflank_F (5′-CTTCTTATC GAT TTTTTATACCGAACATAAAAATAAGGTTAATTA) and K2Lflank_R (5′-CTTCTTCATATGG TGATTGTATTT CCTTGCAATTTAG), and 1024 bp of 3′ homologous sequence (including the native K3L promoter) was amplified with the primers K4Lflank_F (5′-CGTCGTGCGGCCGCCTTGTTAACGGGCTCGTAAATT) and K4Lflank_R (5′-CGA GCGGAGCTCGTACGATACATAGATATTACAAATATCCTAG). A VACV synthetic early/late promoter (SLP) (*Chakrabarti et al., 1997*) was created by annealing primers SLP_F (5′ T CGACAATTGGATCAGCTTTTTTTTTTTTTTTTTTTGGCATATAAATA AGAAGCTTCCCGGGTCTAGAC ) and SLP_R (5′-AGCTCAGATCTGGGCCCTTCG AAGAATAAATATACGGTTTTTTTTTTTTTTTTTC GACTAGGTTAAC). EGFP was amplified from pN1-EGFP (Addgene) using primers EGFP_F (5′-G GAGGACTCGAGATGGTGAGCAAGGGCGA) and EGFP_R (5′-GGAGGTATCGA TTTACTTGTACAG CTCGTCCATGC). Each PCR product was digested with restriction enzymes (New England Biolabs), gel purified (Zymo Research), and sequentially cloned into pB.2 as follows: 5′ flank with ClaI and NdeI, 3′ flank with NotI and SacI, SLP with SalI and XhoI, and EGFP with XhoI and ClaI. The resulting plasmid contained EGFP following SLP, between the two K3L flanking sequences (pB.2-EGFP). The K3L open reading frame was amplified from VC-2 viral DNA using primers K3L_F (5′-GTTGTAG GATCCATGC TTGCATTTTGTTATTCGTTGC) and K3L_R (5′-GTTCTTGTCGACTT ATTGATGTCTACACATCCTTTTG). The eIF2α gene was amplified from human cDNA using primers eIF2α_F (5′-GATGTAGGATCCATGCC GGGTCTAAGTTGTAGAT) and eIF2α_R (5′-CTACTTGTCGACTTAATCTTCAGCTTTGGCTTCCAT). The resulting PCR products were cloned into pB.2-EGFP with BamHI and SalI, placing K3L or eIF2α immediately following the native K3L promoter, and upstream of SLP-EGFP to create pB.2-K3L and pB.2-eIF2α, respectively. Site-directed mutagenesis was performed on pB.2-eIF2α or pB.2-eIF2α-ΔC using primers eIF2α_S51A_F (5′-GATAC GCCTTCTGGCTAATTCACTAAGAAGAATCATGCCTTC) and eIF2α_S51A_R (5′-GAAGGCATGATTCTTCTTAGTGAATTAGCCAGAAGGCGTATC) to generate pB.2-eIF2α-S51A.

Recombinant eIF2α-encoding viruses were constructed by replacing the K3L gene using homologous recombination. RK13-E3K3 cells were infected with VCR2 (MOI=1.0) and transfected at 1 hr post-infection with pB.2-EGFP, pB.2-K3L, pB.2-eIF2α, or pB.2-eIF2α-S51A plasmids by use of FuGENE6 (Promega) according to the manufacturer's protocol. Infected cells were collected at 48 hr post-infection, and viruses were released by one freeze-thaw cycle followed by sonication. Resulting viruses were plaque purified in RK13-E3K3 cells four times, selecting for recombinants expressing EGFP. Final virus clones were verified by PCR and sequencing of viral DNA across the K2L-K4L region of the genome.

## Experimental screen for HGT events

Confluent 150 mm dishes of RK13-K3L cells were infected with VCR2 at an MOI of 0.1. Cell-associated virus was collected after 48 hr of infection, and released from the cell by one freeze-thaw cycle followed by sonication as described (*Isaacs, 2004*). Confluent 150 mm dishes of BHK cells were then infected with ~$10^6$ PFUs of this virus stock for 7 days, during which they were monitored for mCherry expression. About 500 plates were screened, which accounts for an estimated 500 million viruses. For mCherry positive clones observed by fluorescence microscopy, cell-associated virus was collected and released by one freeze-thaw cycle followed by sonication. mCherry-expressing virus was then plaque purified in BHK cells as described (*Isaacs, 2004*).

## Analysis of HGT events by inverse PCR

Virus DNA was extracted from mCherry-expressing clones as previously described from infected BHK cells (*Esposito et al., 1981*). Purified DNA was digested with XbaI and SpeI or BglII (New England Biolabs (NEB)), diluted, and ligated (Quick ligase; NEB) to circularize linear fragments. Primers pointing away from each other in both mCherry (F-CGTGGAACAGTACGAACGCG and R-CCATG TTATCCTCCTCGCCC) and K3L (F-GAGCATAATCCTTCTCGTATACTC and R-GAATATAGGGATAAACT GGTAGGG) were used to amplify regions flanking mCherry-K3L insertions. Resulting PCR bands were

gel purified and Sanger sequenced with the inverse PCR (iPCR) primers. From this sequencing, the general location of the mCherry-K3L insert could be inferred. Primers flanking this region were then used to amplify the putative insertion, along with flanking sequences. These amplicons were then gel purified, Sanger sequenced, and compared to the parent (VCR2) genome to characterize gene integrations. Because isolates 7 and 9 incorporated mCherry-K3L within the repetitive ITR, each end was PCR amplified separately using unique flanking primers and internal primers (mCherry-R and K3L-F, above). However, only the 5′ end of the insertion of isolate 9 was amplified, despite multiple attempts. Thus, we cannot be sure whether this isolate includes a TSD or was cut at a LINE-1 endonuclease site.

## Long read genome sequencing of virus isolates

Virus particles from plaques expressing mCherry-K3L were isolated from BHK cells and virus cores were purified by ultracentrifugation through a 36% sucrose cushion at 60 k rcf for 1 hr. Virus DNA was extracted as previously described (*Esposito et al., 1981*). The SQK-LSK108 library kit (Oxford Nanopore Technologies) was used to prep isolate 1 DNA, which was sequenced on an FLO-MIN106 cell. An SQK-RBK001 kit was used to prep DNA from isolates 2 to 5, which were multiplexed on an FLO-MIN107 cell. Isolates 6–10 were also multiplexed using the SQK-RBK004 library prep kit, and sequenced on FLO-MIN107. Reads were base-called with the Oxford Nanopore Albacore program and aligned to a reference genome that included the VCR2 genome on one contig and the cellularly-expressed mCherry-K3L construct on a separate contig using default NGMLR parameters (*Sedlazeck et al., 2018*). Integrative Genomics Viewer (IGV) software (Broad Institute) was used to visualize insertions (*Robinson et al., 2011*). Sequencing data can be found at the NCBI SRA, accession number: PRJNA614958.

## Measuring titers of virus strains and isolates

We plated $5×10^6$ cells in 100 mm dishes and infected them 16 hr later at an MOI of 0.1. Three plates (biological replicates) were infected with each isolate or strain. After 48 hr, media was aspirated, and cell-associated virus stock was collected by one freeze-thaw and sonication cycle. Then, six-well plates were seeded with $5×10^5$ BHK cells per well, and, 16 hr later, infected with tenfold serial dilutions of virus stock (2 wells per dilution) in 200 µl media. After 2 hr at 37°C, 2 ml of media was added to each well. Cells were fixed and stained with 20% Methanol+0.2% crystal violet 48 hr pos-infection. Wells with 10–100 plaques were counted and averaged to calculate virus titer. Each of the biological replicates is shown in *Figure 2* and *Figure 4*, along with an average of the three and standard deviation.

## Alu assay of RK13 cells

$5×10^5$ HA-HeLa or RK13 cells were plated per 100 mm dish (3× each cell type). After 16 hr, each plate was transfected with 5 µg Alu-Neo (*Dewannieux et al., 2003*). After 2 days, all cells were treated with 2 µg/ml G418. Media were replaced daily for 7 days, after which time it was removed, and cells were fixed and stained with 20% Methanol+0.2% crystal violet.

## Detection of duplication events

Single Nanopore reads with multiple tandem copies of the mCherry-K3L fusion gene were analyzed using IGV (*Robinson et al., 2011*). Duplications were confirmed by PCR, using the iPCR primers in mCherry (F-CGTGGAACAGTACGAACGCG and R-CCATGTTATCCTCCTCGCCC). PCR amplicons were then gel purified and Sanger sequenced to determine break points. However, due to the repetitive nature of the sequence flanking the isolate 7 insertion, the exact break point was not able to be determined.

## Serial passage of VACV-eIF2α virus strains

For each passage, 150 mm dishes were seeded with an aliquot from the same stock RK13 cells ($5×10^6$ cells/dish). For P1, dishes were infected with eIF2α-GFP or eIF2α$^{S51A}$-GFP virus (MOI=1.0) for 2 hr in 5 ml and then supplemented with 15 ml medium. After 48 hr, cells were washed, pelleted, and resuspended in 1 ml of medium. Virus was released by one freeze-thaw cycle followed by sonication. About 900 µl of virus was then used to infect a new dish of cells for P2, and the process was repeated for subsequent passages. Viral titers were determined using the remaining 100 µl of reserved virus stocks from each passage by 48 hr plaque assay in RK13-E3K3 cells.

## Analysis of viral-encoded eIF2α genes

RK13-E3K3 cells were infected with P4 eIF2α-GFP or eIF2α$^{S51A}$-GFP viruses (MOI=0.1) for 24 hr. Virus-infected cells were collected, and total viral DNA extracted as previously described (*Esposito et al., 1981*). The region between K2L and K4L containing the different viral-encoded eIF2α genes was amplified by PCR with primers K2L_seq_F and K4L_seq_R (5'-GGCATTGGTAAATCCTTGCAGA and 5'-CACCTTTTAGTAGGACTAGTATCGTACAA, respectively). SNV detection was performed by sequencing across eIF2α using primers eIF2α_F and eIF2α_R for eIF2α-GFP and eIF2α$^{S51A}$-GFP PCR products, CNV analysis was performed by PCR using primers eIF2α_rep_F (5'-CCTCCTATGGAAGCCAAAGCTGAAGATGAA) and eIF2α_rep_R (5'-CCTCCTATCTACAACTTAGACC CGGCAT) for eIF2α-GFP and eIF2α-$^{S51A}$-GFP viral DNA. Any CNV PCR products formed were sequenced using the same primers for break point detection.

## Acknowledgements

The authors thank D Hancks, E Choung, D Downhour, and E Grow for reagents and advice. This study was supported by NIH grants R35GM134936 (NCE), T32GM007464 (SMF and TAS), T32AI055434 (KRC), and R01AI146915 (SR). NCE was supported by a Burroughs Wellcome Fund Investigator in the Pathogenesis of Infectious Disease award and a HA and Edna Benning Presidential Endowed Chair.

## Additional information

### Competing interests

Nels C Elde: Reviewing editor, eLife. The other authors declare that no competing interests exist.

### Funding

| Funder | Grant reference number | Author |
| --- | --- | --- |
| National Institutes of Health | R35GM134936 | Nels C Elde |
| National Institutes of Health | T32GM007464 | Sarah M Fixsen Thomas A Sasani |
| National Institutes of Health | T32AI055434 | Kelsey R Cone |
| Burroughs Wellcome Fund | 1015462 | Nels C Elde |
| University of Utah | HA and Edna Benning Presidential Endowed Chair | Nels C Elde |
| National Institutes of Health | R01AI146915 | Stefan Rothenburg |

The funders had no role in study design, data collection and interpretation, or the decision to submit the work for publication.

### Author contributions

Sarah M Fixsen, Kelsey R Cone, Data curation, Formal analysis, Validation, Investigation, Visualization, Methodology, Writing - original draft; Stephen A Goldstein, Data curation, Formal analysis, Investigation, Writing – review and editing; Thomas A Sasani, Data curation, Formal analysis, Validation, Visualization; Aaron R Quinlan, Supervision, Funding acquisition, Validation; Stefan Rothenburg, Resources, Funding acquisition, Validation; Nels C Elde, Conceptualization, Resources, Supervision, Funding acquisition, Methodology, Project administration, Writing – review and editing

### Author ORCIDs

Kelsey R Cone http://orcid.org/0000-0002-4547-7174
Thomas A Sasani http://orcid.org/0000-0003-2317-1374
Stefan Rothenburg http://orcid.org/0000-0002-2525-8230
Nels C Elde http://orcid.org/0000-0002-0426-1377

**Decision letter and Author response**
Decision letter https://doi.org/10.7554/eLife.63332.sa1
Author response https://doi.org/10.7554/eLife.63332.sa2

## Additional files

### Supplementary files
• Supplementary file 1. Gene and protein sequence information. Accession numbers for sequences used in phylogenetic analysis shown in *Figure 1B*. Amino acid alignment used in phylogenetic analysis shown in *Figure 1B*. Supplementary sequences.
• Transparent reporting form

### Data availability
Sequencing data have been deposited in the NCBI SRA database under project code PRJNA614958. All data generated or analyses during this study are included in the manuscript and supplementary files.

The following previously published dataset was used:

| Author(s) | Year | Dataset title | Dataset URL | Database and Identifier |
| --- | --- | --- | --- | --- |
| Fixsen SM, Elde NC | 2020 | Vaccinia virus HGT genomes | http://www.ncbi.nlm.nih.gov/sra/PRJNA614958 | NCBI Sequence Read Archive, PRJNA614958 |

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
