## [Editor Report]

This is a very important piece of work and a landmark in understanding the mechanism of horizontal gene transfer between viruses and their hosts. Horizontal gene transfer is a powerful mechanism of rapid evolution in many genomes, including humans and viruses. In this manuscript, a clever selection scheme is implemented to identify poxvirus genomes that have incorporated sequences from their host genome. Subsequent sequence analysis reveals that many of these sequences display the characteristics of terminal sequence repeats, splice junctions like mature mRNAs, and other features characteristics of the movement of LINE elements in cellular DNA genomes. Then, subsequent duplication of these transferred sequences is consistent with the viral DNA replication strategy and rationalizes many of the extant examples of horizontal gene transfer.

---

## [Decision Letter]

**Decision letter after peer review:**

Thank you for submitting your article "Poxviruses capture host genes by LINE-1 retrotransposition" for consideration by *eLife*. Your article has been reviewed by 3 peer reviewers, one of whom is a member of our Board of Reviewing Editors, and the evaluation has been overseen by Patricia Wittkopp as the Senior Editor. The following individuals involved in review of your submission have agreed to reveal their identity: Eugene V. Koonin (Reviewer #2); Derek Walsh (Reviewer #3).

The reviewers have discussed the reviews with one another and the Reviewing Editor has drafted this decision to help you prepare a revised submission.

Summary:

This is a very important piece of work. Horizontal gene transfer is a powerful mechanism of rapid evolution in many genomes, including humans and viruses. In this manuscript, a clever selection scheme is implemented to identify poxvirus genomes that have incorporated sequences from their host genome. Subsequent sequence analysis reveals that many of these sequences display the characteristics of terminal sequence repeats, splice junctions like mature mRNAs and other features characteristics of the movement of LINE elements in cellular DNA genomes. Then, subsequent duplication of these transferred sequences are consistent with the viral DNA replication strategy and rationalize some of the extant examples of horizontal gene transfer.

Essential revisions:

The only requests for modifications come in the presentation. Some of the experimental descriptions and interpretations are written so tersely and vaguely that it is difficult to understand them, let alone critique them. The explicatory parts of this paper are very short and seem to be written for a journal with a less general and curious audience and that enforces restrictive line requirements. This is not the goal of *eLife*. Specific examples of needed rewrites include:

1. In the Introduction, explication is needed for the curious non-expert of the biology and biochemistry of LINE elements and what was known about the mechanisms of their intra-genome movements. Statements such as that on line 67, stating that a target site duplication is consistent with LINE-mediated retrotransposition are not useful in the absence of such an introduction, given that there are many recombination mechanisms that can generate target site duplication. Perhaps that was implied in the phrasing "consistent with". The argument becomes more convincing when, on line 76, sequences consistent with 'a process akin to the generation of processed pseudogenes' are described. It is not until Supplementary Figure 4, and the paragraph that beings on page 114, that the uninitiated reader learns about the LINE endonuclease and reverse transcriptase activities. The biochemical assays for them are not described. It would be much more helpful if this information were in the Intro.

2. The Supplemental Figures contain a wealth of information. Some of this should be in the main manuscript. For example, the genomic structures of the selected viruses – the most important data in the paper – are found in Supplementary Figure 3. At a minimum, these need to be in the main paper, and all the junction sequences in the Supplementary Material.

3. Similarly, the idea to select for the horizontal transfer of eIF2alpha to inhibit PKR is great, and it would be easy to explain to the non-virologist – it actually is, but only in the legend to Supplementary Figure 2. Interesting questions are asked about the ability of both wild-type version of eIF2alpha and a version that is not phosphorylatable being able to rescue a virus with its genes deleted of PKR inhibitors, but the implications for this on the mechanism of eIF2alpha inhibition of PKR are not discussed. Actually, the Supplementary Figures are barely referenced in the main manuscript, and S Figures 1 and 2 not at all.

4. On lines 137-139, there is the interesting line "this activity might enable the persistence of horizontally transferred genes". There are many steps of logic in this suggestion that are interesting and potentially persuasive, and they need to be parsed. It is likely that they are presented in Supplementary Figure 5, which has a six-line legend for a very complicated series of hypotheses.

---

## [Author Response]

Essential revisions:The only requests for modifications come in the presentation. Some of the experimental descriptions and interpretations are written so tersely and vaguely that it is difficult to understand them, let alone critique them. The explicatory parts of this paper are very short and seem to be written for a journal with a less general and curious audience and that enforces restrictive line requirements. This is not the goal of eLife. Specific examples of needed rewrites include:1. In the Introduction, explication is needed for the curious non-expert of the biology and biochemistry of LINE elements and what was known about the mechanisms of their intra-genome movements. Statements such as that on line 67, stating that a target site duplication is consistent with LINE-mediated retrotransposition are not useful in the absence of such an introduction, given that there are many recombination mechanisms that can generate target site duplication. Perhaps that was implied in the phrasing "consistent with". The argument becomes more convincing when, on line 76, sequences consistent with 'a process akin to the generation of processed pseudogenes' are described. It is not until Supplementary Figure 4, and the paragraph that beings on page 114, that the uninitiated reader learns about the LINE endonuclease and reverse transcriptase activities. The biochemical assays for them are not described. It would be much more helpful if this information were in the Intro.

We thank the reviewers for pointing out issues in the introduction and present a reworked and expanded introduction that introduces LINE-1 biology, defines genetic hallmarks of LINE-1 activity, and sets up the study with a broader audience in mind.

2. The Supplemental Figures contain a wealth of information. Some of this should be in the main manuscript. For example, the genomic structures of the selected viruses – the most important data in the paper – are found in Supplementary Figure 3. At a minimum, these need to be in the main paper, and all the junction sequences in the Supplementary Material.

We promoted Supplemental Figure 3 to Figure 3B and swapped the details related to junction sequences to Supplemental Figure 3 to better emphasize the most important findings. We also split Figure 4 into two figures, with new Figure 5 displaying the model of horizontal gene transfer to better tie together the whole manuscript.

3. Similarly, the idea to select for the horizontal transfer of eIF2alpha to inhibit PKR is great, and it would be easy to explain to the non-virologist – it actually is, but only in the legend to Supplementary Figure 2. Interesting questions are asked about the ability of both wild-type version of eIF2alpha and a version that is not phosphorylatable being able to rescue a virus with its genes deleted of PKR inhibitors, but the implications for this on the mechanism of eIF2alpha inhibition of PKR are not discussed. Actually, the Supplementary Figures are barely referenced in the main manuscript, and S Figures 1 and 2 not at all.

Thank you to the reviewers for pointing out this opportunity to better set up and explain the experiments with viruses expressing eIF2alpha and for alerting us to improve connections to the supplemental materials. We have revised the Results section to better explain how ancient horizontal transfers of the eIF2alpha gene evolved to become K3L in modern poxviruses. We now also refer to all the supplemental figures in the main text.

4. On lines 137-139, there is the interesting line "this activity might enable the persistence of horizontally transferred genes". There are many steps of logic in this suggestion that are interesting and potentially persuasive, and they need to be parsed. It is likely that they are presented in Supplementary Figure 5, which has a six-line legend for a very complicated series of hypotheses.

We have expanded our thinking in this part of the discussion and linked it to Figure 5 – supplemental figure 1, where the models are now expanded on as well as in the legend.